# Circulating Insulin-Like Growth Factor I is Involved in the Effect of High Fat Diet on Peripheral Amyloid β Clearance

**DOI:** 10.3390/ijms21249675

**Published:** 2020-12-18

**Authors:** Raquel Herrero-Labrador, Angel Trueba-Saiz, Laura Martinez-Rachadell, Mᵃ Estrella Fernandez de Sevilla, Jonathan A. Zegarra-Valdivia, Jaime Pignatelli, Sonia Diaz-Pacheco, Ana M. Fernandez, Ignacio Torres Aleman

**Affiliations:** 1Cajal Institute, CSIC, 28002 Madrid, Spain; rhlabrador@cajal.csic.es (R.H.-L.); angelts86@gmail.com (A.T.-S.); rachadell@cajal.csic.es (L.M.-R.); mefernandezdesevilla@cajal.csic.es (M.E.F.d.S.); zegarra@cajal.csic.es (J.A.Z.-V.); pigna@cajal.csic.es (J.P.); soniadiaz@cajal.csic.es (S.D.-P.); anaf@cajal.csic.es (A.M.F.); 2Ciberned, 28029 Madrid, Spain; 3Universidad Nacional de San Agustín de Arequipa, 04001 Arequipa, Peru

**Keywords:** diet, Aβ clearance, IGF-I, Alzheimer disease

## Abstract

Obesity is a risk factor for Alzheimer’s disease (AD), but underlying mechanisms are not clear. We analyzed peripheral clearance of amyloid β (Aβ) in overweight mice because its systemic elimination may impact brain Aβ load, a major landmark of AD pathology. We also analyzed whether circulating insulin-like growth factor I (IGF-I) intervenes in the effects of overweight as this growth factor modulates brain Aβ clearance and is increased in the serum of overweight mice. Overweight mice showed increased Aβ accumulation by the liver, the major site of elimination of systemic Aβ, but unaltered brain Aβ levels. We also found that Aβ accumulation by hepatocytes is stimulated by IGF-I, and that mice with low serum IGF-I levels show reduced liver Aβ accumulation—ameliorated by IGF-I administration, and unchanged brain Aβ levels. In the brain, IGF-I favored the association of its receptor (IGF-IR) with the Aβ precursor protein (APP), and at the same time, stimulated non-amyloidogenic processing of APP in astrocytes, as indicated by an increased sAPPα/sAPPβ ratio after IGF-I treatment. Since serum IGF-I enters into the brain in an activity-dependent manner, we analyzed in overweight mice the effect of brain activation by environmental enrichment (EE) on brain IGF-IR phosphorylation and its association to APP, as a readout of IGF-I activity. After EE, significantly reduced brain IGF-IR phosphorylation and APP/IGF-IR association were found in overweight mice as compared to lean controls. Collectively, these results indicate that a high-fat diet influences peripheral clearance of Aβ without affecting brain Aβ load. Increased serum IGF-I likely contributes to enhanced peripheral Aβ clearance in overweight mice, without affecting brain Aβ load probably because its brain entrance is reduced.

## 1. Introduction

Obesity is considered a risk factor for AD [1,2,3]. However, the relationship between body weight and dementia appears complex [4,5,6], and recent observations even pose a protective role of late-life excess weight in AD [7]. Taking into account the worrying worldwide prevalence of obesity and dementia [8,9], greater knowledge of possible links between the two conditions is imperative. Amyloid β (Aβ) handling may be one such link, as this peptide is considered a major pathogenic factor in AD and obesity-associated inflammation [10] may interfere with its elimination from the brain.

In turn, we recently proposed that insulin peptides such as insulin and insulin-like growth factor I (IGF-I), may be involved in the connection between lifestyle and AD risk [11], although apparently, contradictory evidence links IGF-I with AD. Thus, circulating IGF-I has been reported to favor brain Aβ clearance [12], while IGF-IR seems to interfere with proteostasis, favoring brain accumulation of Aβ [13]. Of note, hepatocytes, the main source of circulating IGF-I [14], are the major disposal system for circulating Aβ in mice [15], and previous evidence has shown that insulin, a hormone closely related to IGF-I, favors Aβ uptake by hepatocytes [16].

IGF-I may not only be involved in brain Aβ clearance but also in its production. Accordingly, an effect of IGF-I was observed on APP processing towards the non-amyloidogenic pathway, reducing in this way its production [17,18,19]. However, IGF-I has also been shown to stimulate the amyloidogenic pathway [20,21]. Thus, the actions of IGF-I on Aβ production are not yet clear either.

Significantly, the actions of circulating IGF-I in the brain are modulated by diet [22]. Brain IGF-I is in part locally synthesized [23], and in part derived from uptake from the circulation [24]. The entrance of circulating IGF-I into the brain is activity-dependent and tightly regulated [25], probably because it participates in many essential brain functions [26]. Furthermore, the entrance of serum IGF-I into the brain may be altered in pathological conditions, as seen by us in stressed animals [11].

In the present work, we investigated the regulation of peripheral Aβ clearance in overweight mice and the possible role of circulating IGF-I.

## 2. Results

### 2.1. High Fat Diet Influences Peripheral Aβ Disposal and Serum IGF-I Levels

We examined plasma Aβ clearance through liver uptake using fluorescent Aβ_1-40_ [27]. While Aβ_1-42_ is an important component of brain Aβ plaques, Aβ_1-40_ is the major form of soluble, circulating Aβ. The liver is the major systemic disposal route of Aβ [15], and peripheral Aβ disposal is a proposed mechanism for central Aβ clearance [28]. We injected fluorescently tagged Aβ_1-40_ through the tail vein of overweight mice under high-fat diet (HFD) and their lean controls receiving a standard diet. Ninety minutes later, animals were culled, and fluorescence levels were examined in the liver and plasma. Compared to controls, overweight mice showed significantly increased fluorescence accumulation in the liver and decreased in serum, suggesting increased uptake of fluorescent Aβ_1-40_ by hepatocytes, leading to its increased clearance in serum (Figure 1A). Since circulating IGF-I may affect Aβ clearance [12], we examined serum IGF-I levels in overweight mice and found them increased (Figure 1B). However, when we measured endogenous soluble brain Aβ_1-40_ levels with a sensitive mouse-specific ELISA that detects its monomeric forms, similar Aβ_1-40_ levels were found in intact overweight mice, as compared to control lean mice (Figure 1C). Since serum IGF-I levels are increased in overweight mice (Figure 1B), we determined whether brain IGF-I is correspondingly higher, as serum IGF-I crosses the BBB [24]. However, overweight mice showed normal brain IGF-I levels (Figure 1D).

### 2.2. IGF-I Promotes Aβ Uptake by Hepatocytes

To try to clarify the discrepancy between increased peripheral disposal of Aβ and normal brain load, we analyzed a possible role of IGF-I in peripheral Aβ clearance, which predominantly takes place in the liver [14]. As brain cells take up Aβ to clear it [29], we determined whether IGF-I modulates in vitro uptake by hepatocytes of fluorescently labeled Aβ, as an indirect measure of its potential effects on Aβ disposal by the liver. In the presence of IGF-I (10 nM), hepatocytes accumulated significantly more fluorescence, suggesting a stimulatory action of IGF-I on Aβ uptake by these cells (Figure 2A). Moreover, liver IGF-I deficient (LID) mice with a 70% reduction in circulating IGF-I [30] showed reduced liver accumulation of tagged Aβ after intravenous injection, while blood levels were increased, as compared to controls, indicating reduced liver clearance (Figure 2B). Of note, IGF-I treatment of LID mice ameliorated these deficits (Figure 2B). However, as in overweight mice, LID mice did not show changes in brain Aβ_1-40_ levels (Figure 2C).

### 2.3. Cell-Specific Actions of IGF-I in APP Metabolism by Brain Cells

IGF-I has been reported to promote either amyloidogenic [20], or non-amyloidogenic [18] APP processing pathways in neuronal cell lines. To clarify its role in primary cells, we analyzed the actions of IGF-I on amyloidogenic and non-amyloidogenic APP processing by astrocytes and neurons, the primary sources of Aβ in the brain [31,32]. Using the soluble APP metabolites sAPPβ and sAPPα as markers of the amyloidogenic and the non-amyloidogenic pathway, respectively, we found that IGF-I modulates their production in a cell-specific fashion. In astrocytes, secretion of both soluble forms of APP was stimulated by IGF-I, whereas in neurons, IGF-I inhibited their secretion (Figure 3A). However, the APPα/sAPPβ ratio was increased in both cell types, indicating that the net action of IGF-I is to promote the non-amyloidogenic processing of APP (Figure 3B).

Since both IGF-IR and APP associate to LRP1 and APP processing depends on its subcellular localization [33], we assessed whether IGF-IR and APP interact with each other. Indeed, IGF-IR and APP co-immunoprecipitated in astrocytes, whereas in neurons, the interaction was negligible (Figure 3C). Proximity ligation assays (PLA) confirmed a robust interaction of APP with IGF-IR in astrocytes (Figure 3D), while in neurons, the interaction was negligible (not shown). Treatment of astrocytes with IGF-I resulted in a significantly increased interaction between both proteins, as determined by a stronger PLA signal (Figure 3D).

Since IGF-I promotes Aβ uptake by hepatocytes [34], we examined whether it exerts similar action in brain cells. In this organ, the main cell types involved in Aβ clearance are microglia and astrocytes through its uptake and degradation [35,36] and endothelial cells at the blood-brain-barrier (BBB) through efflux of brain Aβ into the circulation [37]. We found that IGF-I promoted Aβ uptake by astrocytes (Figure 4A) while it decreased it in microglia (Figure 4B). In brain endothelial cell cultures mimicking the BBB architecture [25], IGF-I did not significantly affect Aβ efflux from the “brain” side to the “blood” side of the double chamber, although it was slightly reduced (Figure 4C).

### 2.4. Reduced Brain IGF-I Activity in Overweight Mice

To tried to explain the discrepancy between peripheral and central IGF-I levels in overweight mice, we determined whether the passage of serum IGF-I into the brain is reduced in them. To this end, we took advantage that exposure to environmental enrichment (EE) stimulates the passage of serum IGF-I into the brain in normal mice [25]. We tested whether overweight mice would show an altered passage of IGF-I after EE by measuring Tyr-phosphorylation of brain IGF-I receptors as a proxy of their activity. While basal levels of brain Tyr phosphorylation of IGF-IR were similar in control and overweight mice, after EE, the latter showed reduced IGF-IR phosphorylation (Figure 5A), pointing to the impaired entrance of circulating IGF-I. In addition, we used the interaction of IGF-IR with APP seen in vitro as an additional indicator of the entrance of IGF-I into the brain of EE-stimulated overweight mice. Significantly, whereas in lean mice, EE produced enhanced brain APP/IGF-IR interactions, in overweight mice, this interaction was significantly smaller (Figure 5B), pointing to the reduced entrance of IGF-I.

## 3. Discussion

These results suggest that in adult normal mice, a high fat diet influences plasma Aβ clearance through the liver, and that circulating IGF-I may play a role in this effect. Of note, no correlation was found between increases or decreases in peripheral Aβ clearance and brain Aβ levels. These observations agree with previous ones showing that reducing peripheral Aβ does not affect brain Aβ levels or that there is no correlation between central and peripheral Aβ levels in AD patients [38,39,40,41]. Compartmentalization of Aβ clearance may be reflecting multiple sources of this circulating peptide, as under normal physiological circumstances, Aβ is produced not only in the brain but throughout the body.

However, other observations do not support the compartmentalization of Aβ clearance. Thus, increased peripheral Aβ levels after anti-Aβ treatment were reported to parallel a decrease of brain Aβ; reducing peripheral Aβ was sufficient to reduced brain Aβ, and recent studies favor a diagnostic utility of the relationship between plasma and CSF Aβ_1-42_ [28,42,43]. Thus, the relationship between peripheral and central Aβ is still under debate [44]; indeed, a substantial part of brain Aβ clearance in humans takes place in the periphery [45], and reduction of peripheral Aβ reduces its content in the CSF [46].

In turn, normal brain levels of Aβ in overweight mice agree with previous observations [47], although increased brain Aβ load in overweight mice was found by others [48,49]. Conversely, enhanced elimination of circulating Aβ in overweight mice favors the notion that a higher body mass index may be protective rather than detrimental for AD risk [4]. Although counter-intuitive, the normal brain Aβ load seen by us in overweight mice may fit with this notion, as a HFD may produce higher brain Aβ levels that could be cleared faster and, therefore, no changes would be detected. In addition, other still undefined systemic changes may contribute to brain Aβ load in AD, as recently postulated [50].

Among the latter, we considered circulating IGF-I as a probable systemic factor influencing AD. Indeed, there are many reports indicating a relationship of circulating IGF-I with the disease, although no consensus has been reached on whether it exerts a beneficial or detrimental action [51], or could serve as a biomarker of the evolution of AD [52,53]. The initial observation that IGF-I is involved in brain Aβ clearance [12]—although this has been questioned [54], supported these studies. In relation to the present findings, IGF-I is known to show diet-sensitive actions in the brain [22]. Further, previous studies using the APP/PS1 mouse model of AD had shown that HFD interferes with central and peripheral insulin signaling [55], which is largely shared by IGF-I. These authors also found impaired insulin signaling in wild-type mice submitted to HFD, which agrees with our present findings. Hence, several observations favor involvement of impaired activity of circulating IGF-I in the actions of a high-fat diet on peripheral Aβ clearance. (1) IGF-I levels are increased in overweight mice, (2) IGF-I, as previously seen with insulin [16], stimulates the uptake of Aβ by hepatocytes, and (3) LID mice with low serum IGF-I show reduced peripheral Aβ clearance that was ameliorated by treatment with systemic IGF-I. Thus, higher serum IGF-I levels in overweight mice may contribute to enhance plasma Aβ clearance without affecting brain Aβ load. The latter disagrees with our previous observation of increased brain Aβ levels in LID mice [12]. The use of an in-house ELISA and formic acid extraction of total brain Aβ (prior results) vs. a commercial ELISA with guanidinium HCl extraction of brain Aβ (current results), and/or changing housing conditions of LID mice over time in our animal facility (i.e., sterile food pellets) affecting their microbiome [56], that shows strong interactions with IGF-I function [57], may explain this discrepancy. However, we do not have a straightforward explanation of this important difference with our previous results.

As indicated by an increased sAPPα/sAPPβ ratio in IGF-I-treated neurons and astrocytes, the net action of IGF-I on the main cell types producing Aβ in the brain is to increase non-amyloidogenic processing of APP, contributing in this way to lower its brain levels and enhance neuroprotection, as sAPPα is neuroprotective acting in part through IGF-IR [58]. Thus, the overall action of IGF-I in the brain may be anti-amyloidogenic. Further studies using the sAPPα/sAPPβ ratio in brain samples will help confirm this observation. Intriguingly, insulin favors Aβ secretion in neurons [59], suggesting a complex interplay of these hormones in regulating brain Aβ levels. At the same time, reduced IGF-I entrance in the brain of overweight mice may hamper its anti-amyloidogenic actions. Indeed, overweight mice showed not only reduced entrance of serum IGF-I in response to EE stimulation, as determined by reduced brain IGF-IR phosphorylation but also reduced APP/IGF-IR interaction. In previous work, we documented an inhibitory effect of triglycerides (TGLs) in BBB entrance of IGF-I across the choroid plexus [22]. It is possible that high serum TGLs as a result of the high-fat diet also interferes with the BBB entrance of IGF-I across brain endothelial cells in overweight mice, as previously seen for other circulating hormones such as insulin [60] or leptin [61].

Reduced IGF-I entrance would affect its pro-clearance actions on brain Aβ [12,62]. In addition, we cannot discard that the inhibitory actions of IGF-I on Aβ uptake by microglia may also counteract its actions on astrocytes (but see below). Alternatively, brain Aβ levels may not be affected by peripheral Aβ clearance, or other factors may also contribute to it, such as the recently postulated vascular drainage [63,64]. Interestingly, insulin also enhances the degradation of Aβ and its clearance in astrocytes [65] and hepatocytes [16], respectively. Thus, these two closely related hormones may modulate Aβ disposal in a concerted manner, as previously reported for glucose handling [66].

IGF-I stimulates Aβ uptake by astrocytes, while inhibits it in microglia. Whereas, astrocytes appear critical to determine Aβ load [67], and increased clearance of Aβ by astrocytes may result in reduced Aβ plaques [36], inhibition of Aβ uptake by microglia may also reduce plaques [68], as the role of microglial uptake of Aβ in plaque formation may be detrimental [69,70]. In accordance with a stimulatory effect of IGF-I on astrocytes, previous observations suggested that astrocyte-derived IGF-I protects neurons against Aβ toxicity through a mechanism involving its uptake [71].

The observed astrocyte-specific interaction of APP with IGF-IR and on sAPPα and sAPPβ levels may be related to a differential processing of APP by IGF-I in these cells since its processing depends on its intracellular localization [33]. Of the different isoforms of APP, the major one expressed in neurons is APP_695_, which lacks the extracellular Kunitz protease inhibitor (KPI) protein–protein interaction domain KPI. This domain is present in the longer isoforms, APP_751_ and APP_770_, which are the most abundant types in glial cells [72]. It is possible that KPI is involved in the observed interactions with IGF-IR in astrocytes, but this requires further analysis. In turn, a trend of IGF-I to inhibit brain efflux of Aβ through BBB endothelial cells would favor its accumulation in the brain parenchyma [73]. We previously reported that IGF-I stimulates Aβ efflux through the choroid plexus BBB [12], an observation supported by the reducing effects of in vivo IGF-I administration on brain Aβ levels [12,62]. Thus, IGF-I may show site-specific effects on Aβ efflux through BBB cells.

Several limitations of this study need to be pointed out. We determined plasma Aβ clearance using exogenously added tagged Aβ. Measuring dynamic changes in circulating levels of endogenous Aβ in overweight mice would be necessary to firmly establish that peripheral Aβ clearance is enhanced. However, available methods of quantification of serum Aβ are not sensitive enough to reliably detect decreases in non-transgenic mice. Further, since other organs also accumulate and degrade circulating Aβ, they may also contribute to diet influence on plasma Aβ clearance. In addition, whether diet influences clearance of other circulating peptides needs to be clarified. Finally, mice with low serum IGF-I (LID mice) present insulin resistance [74], which may affect the ability of hepatocytes to take up and degrade Aβ.

## 4. Materials and Methods

### 4.1. Materials

Human IGF-I was purchased from PeProTec (London, UK). Primary antibodies were monoclonal anti-IGF-I receptor (1:1000; Santa Cruz Biotechnology, Dallas, TX, USA), monoclonal anti-APP (Nt 22C11; Millipore; 1:200), for PLA studies, polyclonal anti-APP (1:200; Sigma, Madrid, Spain), for immunoprecipitation, and monoclonal anti-pTyr (1:1000, Transduction Labs, BD BioSciences, San Jose, CA, USA). Secondary antibodies were goat anti-rabbit (1:20,000) or mouse IRDye-coupled (1:20,000), both from LI-COR (Lincoln, NE, USA).

### 4.2. Animals

New-born wild type (WT) C57BL6/J mice were used for cell cultures, and WT and liver IGF-I deficient (LID mice; bred in-house, congenic with C57/BL6/J) male adult mice (3–5 months old) were used for the rest of the experiments. LID mice presented low levels of serum IGF-I due to the disruption of the liver *IGF-I* gene with the albumin-Cre/Lox system [14]. Serum IGF-I deficient mice have normal body and brain weights, and they do not show any major developmental defects [14,75]. In rescuing experiments, LID mice were treated with subcutaneous hIGF-I (5 µg/kg/day) for 30 days, using osmotic pumps (1004 Alzet, Durect Corp, Cupertino, CA, USA) following the manufacturer’s instructions. Animal procedures followed the European (86/609/EEC and 2003/65/EC, European Council Directives) and approval of the local Bioethics Committee.

### 4.3. High Fat Diet

Wild type C57BL6/J mice were fed for 10 weeks with either a control diet (ref E15000-04), or a high-fat diet (HFD) with 45% KJ fat + 1.25% cholesterol (ref E15744-34), both purchased from ssniff Spezialdiäten GmbH (Soest, Germany). After 10 weeks, animals were overweight (Appendix A), developed glucose intolerance (Appendix A), together with hyperinsulinemia and insulin resistance (not shown).

### 4.4. Cell Cultures

Astroglial cultures with >95% GFAP-positive cells were prepared as described [76]. Postnatal (day 1–2) brains were dissected, the forebrain removed, and mechanically dissociated. The resulting mixed cell suspension was centrifuged and plated in DMEM/F-12 (Life Technologies, Carlsbad, CA, USA) with 10% fetal bovine serum (Life Technologies) and 100 mg/mL of antibiotic-antimycotic solution (Sigma-Aldrich, Madrid, Spain). When confluent, cells were shaken (210 rpm/37 °C/3 h) to detach microglial cells. For microglial cultures, supernatants were centrifuged (1000 rpm/5 min), re-suspended in DMEM/F12 (Life Technologies) + FBS (Gibco, Gaithersburg, MD, USA), HS, and penicillin/streptomycin solution. Cells were seeded at 12.5 × 10^4^ cells/cm^2^ in a multi-well coated with poly-L-lysine [77] and cultured for 2 days. Cells were then changed to DMEM/F12 for 3 h until Aβ uptake was carried out (see below). Astrocytes were then collected from the same flasks that microglia was obtained, as follows. After removing the microglia-containing supernatant, the medium was replaced, and flasks were shaken for 15 h/280 rpm. Cells were then trypsinized and seeded at 3.75 × 10^4^ cells/cm^2^ in the same culture medium, replaced every 4 days. When 80% confluency was reached, astrocytes were cultured for 3 h with DMEM/F12 before the different assays were initiated (see below). Endothelial cell cultures were performed as described [77]. Briefly, dissection was performed on ice, and cortices were cut into small pieces (1 mm^3^), digested in a mixture of collagenase/dispase (270 U collagenase/mL, 10% dispase) and DNAse (10 U/mL) in DMEM for 1.5 h at 37 °C. The cell pellet was separated by centrifugation in 20% bovine serum albumin/DMEM (1000× *g*, 5 min). Capillary fragments were retained on a 10 µm nylon filter, removed from the filter with endothelial cell basal medium (Life Technologies, Waltham, MA, USA), supplemented with 20% bovine plasma-derived serum and antibiotics (penicillin, 100 U/mL; streptomycin, 100 µg/mL), and seeded on 60 mm Petri dishes multi-well plate coated with collagen type IV (5 µg/cm^2^) and fibronectin (1 µg/cm^2^). 3 µg/mL puromycin was added for 3 days, removed from the culture medium, and replaced by fibroblast growth factor (2 ng/mL) and hydrocortisone (1 µg/mL). For hepatocytes cultures, adult (2 months old) control animals were anesthesized (pentobarbital 50 mg/kg), and the hepatic portal vein exposed to inject a solution containing NaCl (118 Mm), KCl (4.7 Mm), KH_2_PO_4_ (1.2 Mm), NaHCO_3_ (25 Mm), glucose (5.5 Mm), and EGTA (0.5 Mm) at 37C. The inferior cava vein was cut to open the circuit. Thereafter the same solution without EGTA and containing CaCl_2_ (2 Mm), MgSO_4_ (1.2 Mm), and collagenase (90 U/mL) were perfused. The liver was dissected and placed in DMEM/F12-10% FBS with penicillin/streptomycin, filtered in a 70 um Nylon mesh, centrifuged (60× *g*, 5 min), and re-suspended in DMEM/F12-10% FBS with 45% Percoll^®^ (Sigma Aldrich). Cells were then re-suspended and washed 3X in DMEM/F12-10% FBS using 200× *g*, 10 min spins, before plating them at 8.25 × 10^4^ cells/collagen-coated multi-well. Cultures were kept 2 days before use.

### 4.5. Glucose Tolerance Test (GTT)

Mice were fasted for 6 h and left isolated in individual cages (with water but no food access) for at least 30 min before starting the test to avoid any stress-related effect on glycemia [78]. For the glucose tolerance test (GTT), an overload of glucose (2 g/kg) was injected intraperitoneally. The aqueous solution was left overnight at room temperature, thus the β-form of glucose was enriched. Blood samples were extracted from the tip of the tail at times 0, 15, 30, 60, and 90 to measure glucose levels with a glucometer (Menarini Diagnosis, Florence, Italy).

### 4.6. Environmental Enrichment

Mice were submitted to environmental enrichment, as explained in detail elsewhere [79]. Briefly, animals were placed for 2 h in a large cage, 10 animals/cage, and with different objects (cardboard tunnels, shelters of different materials, a plastic net, toys, chewable, and nesting material). Thereafter, they were sacrificed, and their brain collected for immunoprecipitation and western blot analysis.

### 4.7. Aβ Uptake

Aβ uptake was used as a proxy of its clearance as cells take it up to subsequently degrade it.

In vitro: Cells were treated during 15 h with 500 nM soluble Aβ40-HiLyte Fluor™ 488 (AnaSpec, San Jose, CA, USA) [80], and IGF-I (1 nM in glial cultures, 10 nM in hepatocytes). Thereafter, cultures were washed with PBS pH 6.0 to eliminate membrane-bound Aβ followed by PBS pH 7.4. Cell nuclei were stained with Hoechst 33,342 (Thermo Fisher Scientific, Waltham, MA, USA; 1:500) in PBS pH 7.4/5 min, fluorescent images were taken in a DMI 6000 (Leica) microscope using Exc: 350 nm/Em: 461 nm for Hoeschst dye and Exc: 503 nm/Em: 528 nm for fluorescently labeled Aβ. Thereafter, cells were lysed in Tris-HCl (10 mM) pH 8.0, guanidine (50 mM), and spun at 14.000 rpm for 10 min at 4 °C. Fluorescence was quantified in a FLUOStar OPTIMA (BMG Labtech, Thermo Fisher) at Exc: 485 nm/Em: 520 nm. In transcytosis assays using brain endothelial cells, Aβ40-HiLyte Fluor™ 488 soluble (500 nM) was added in the bottom compartment (Figure 4C) with or without 1 nM IGF-I, and after 15 h the culture medium from the upper chamber was collected and fluorescence measured in the fluorimeter, as above.

In vivo: Aβ40-HiLyte Fluor™ 488 (400 μg/kg) was injected into the tail vein using a 0.38 mm cannula (Intramedic, Madrid, Spain), and after 90 min mice were sacrificed, blood taken from the heart and liver dissected. This timing was chosen based on previous observations of peak plasma Aβ clearance [27]. Liver tissue was homogenized in Tris-HCl (10 mM) pH 8.0-guanidine (50 mM). Fluorescence in serum and liver extracts was quantified by fluorimetry, as above. Values were normalized per ml of serum or mg of protein. The latter was measured in liver samples using the BCA system (Sigma).

### 4.8. Immunoassays

Immunoprecipitation (IP) was performed as described before [81]. In brief, cultured cells or brain tissue were homogenized in ice-cold buffer with 10 mM Tris HCl pH 7.5, 150 mM NaCl, 1 mM EDTA, 1 mM EGTA, 1% Triton X-100, 0.5% NP40, 1 mM sodium orthovanadate, and a protease inhibitor cocktail (Sigma) plus 2 mM PMSF, using 1 mL of buffer per mg of tissue. Insoluble material was removed by centrifugation, and supernatants were incubated overnight at 4 °C with the antibodies: Monoclonal anti-APP (22C11, Millipore, Watford, UK), monoclonal anti-pTyr (PY20, BD Transduction Labs), or polyclonal anti-IGF-IR (D23H3, Cell Signalling, Danvers, MA, USA). Immunocomplexes were collected with Protein A/G agarose (Santa Cruz Biotechnology, Sta Cruz, CA, USA) for 1 h at 4 °C and washed 3X in homogenization buffer before separation by SDS-polyacrylamide gel electrophoresis and transferred to nitrocellulose membranes. After blocking for 1 h with 5% BSA in TTBS (20 mM Tris-HCl, pH 7.4, 150 M NaCl, 0.1% Tween 20), membranes were incubated overnight at 4 °C with the different antibodies in TTBS, washed, incubated with secondary antibodies and develop using the Odissey procedure (Lycor Biosciences, Lincoln, USA). Immunoprecipitates with non-immune IgG and total lysates (input) were blotted with anti-IGFIR as controls (see Appendix A). A representative IP blot is shown from a total of at least 3 independent experiments. GFAP immunocytochemistry in cultured cells followed previously published procedures [81]. In brief, cultured cells were incubated to block non-specific antibody binding, followed by incubation overnight at 4 °C with anti-GFAP in phosphate buffer (PB)—1% bovine albumin—1% Triton X-100 (PBT). After several washes in PB, sections were incubated with an Alexa-coupled secondary antibody (1:1000, Molecular Probes, Eugene, USA) diluted in PBT. Finally, a 1:500 dilution (in PBS) of DAPI (Hoechst 33342) was added for 3 min. Wells were rinsed several times in PB 0.1 N, mounted with 15 µL of gerbatol mounting medium, and allowed to dry. The omission of the primary antibody was used as a control. Microphotographs were taken in a Leica (Wetzlar, Germany) microscope.

IGF-I in serum and brain was determined using a species-specific ELISA (R&D Systems, Minneapolis, USA), as described in detail elsewhere [26]. Murine Aβ (Thermo Fisher Scientific) was determined by ELISA in brain lysates, following the manufacturer´s instructions. This commercial system rules out background signal due to non-APP proteins, as indicated elsewhere using brain lysates from APP knock out mice [82]. Brain Aβ was extracted by tissue homogenization in 200 mM guanidine-HCl, 20 mM Tris-HCl, pH 8.0 with protease inhibitors. Then, the homogenate was mixed for 3 h, centrifuged at 15,000 × *g* for 20 min at 4 °C, and the supernatant diluted two-fold with Standard diluent buffer included in the kit. The ELISA procedure was carried out following the manufacturer´s instructions. Murine sAPPα and sAPPβ were determined by ELISA in culture supernatants. Brain samples were normalized with total protein determined by BCA method (Sigma-Aldrich). Blood was collected from the heart after pentobarbital anesthesia and thereafter brains were dissected and frozen at −80 °C until used.

### 4.9. Proximity Ligation Assays (PLA)

Assays were performed as described [83]. Amyloid precursor protein (APP)—IGF-IR interactions were detected in astrocytes and neurons grown on glass coverslips using the Duolink II in situ PLA detection Kit (OLink; Bioscience, Sweden). Cultured cells were fixed in 4% paraformaldehyde/10 min, washed with PBS containing 20 mM glycine to quench the aldehyde groups, permeabilized with the same buffer containing 0.05% Triton X-100 for 5 min, and washed with PBS. After 1 h/37 °C with the blocking solution in a pre-heated humidity chamber, cells were incubated overnight in antibody diluent medium with primary antibodies: Mouse monoclonal anti-APP and rabbit polyclonal anti-IGF-I receptor, and processed following the instructions of the supplier using the PLA probes detecting rabbit or mouse antibodies (Duolink II PLA probe anti-Rabbit plus, and Duolink II PLA probe anti-Mouse minus, diluted 1:5 in antibody diluent), and a DAPI-containing mounting medium.

### 4.10. Statistical Analysis

Data were analyzed with GraphPad Prism 6.0 software. A normal distribution Kolmogorov–Smirnov test was carried out in all experiments and a non-parametric Wilcoxon test was applied accordingly. For samples with normal distribution, parametric tests include one or two-way ANOVA followed by a Bonferroni or *t*-test. A *p* < 0.05 was considered significant.

## 5. Conclusions

In summary, a high fat diet influences peripheral Aβ clearance. A lack of correlation between peripheral clearance and central Aβ load further supports a non-linear relationship between both compartments. Actions of IGF-I on Aβ handling may be related to diet influences on AD pathology; therefore, cellular sites of IGF-I interaction may constitute new druggable targets, through, for example, potentiating the passage of circulating IGF-I into the brain across the BBB. However, caution is needed as previous attempts to exploit IGF-I as a therapeutic effector in AD have been unfruitful [84].

## Figures and Tables

**Figure 1 ijms-21-09675-f001:**
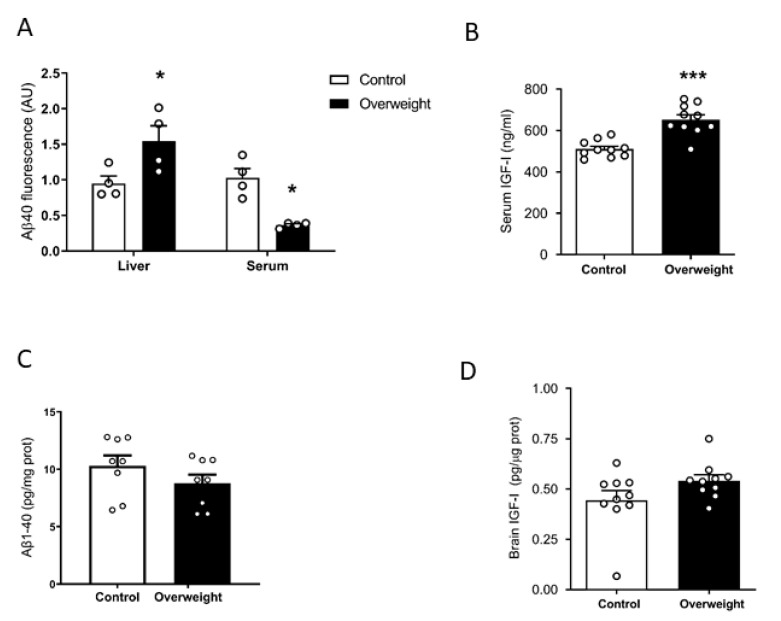
Influence of high fat diet on Aβ and IGF-I levels. (**A**) Overweight mice show significantly increased fluorescence in the liver and reduced in serum, suggesting increased uptake of fluorescently labeled Aβ_1-40_ by the liver and increased clearance from serum (*n* = 4 per group). (**B**) Serum levels of IGF-I are increased after 10 weeks of a high-fat diet (*n* = 10 per group). (**C**) Brain Aβ_1-40_ levels remain unaltered in overweight mice as compared to lean controls (*n* = 6–8). * *p* < 0.05 and *** *p* < 0.001. (**D**) Brain levels of IGF-I were normal in high fat diet (HFD)-fed overweight mice (*n* = 10 per group).

**Figure 2 ijms-21-09675-f002:**
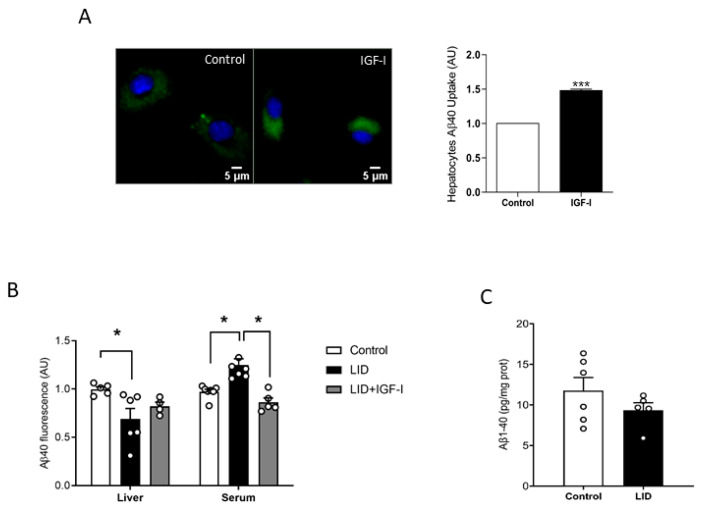
Modulation by IGF-I of Aβ uptake by hepatocytes. (**A**) IGF-I induces uptake of Aβ by hepatocytes (*n* = 6). Representative micrograph of cultured hepatocytes with internalized fluorescent Aβ (green). Cell nuclei stained with Hoescht. Lower histograms: Quantification of intracellular fluorescent Aβ after IGF-I treatment. (**B**) Serum IGF-I deficient mice (liver IGF-I deficient (LID) mice) show reduced Aβ uptake by the liver, which was ameliorated by systemic IGF-I treatment (*n* = 5 control/6 LID/5 LID + IGF-I). (**C**) Brain Aβ levels did not change in LID mice (*n* = 8). * *p* < 0.05 and *** *p* < 0.001.

**Figure 3 ijms-21-09675-f003:**
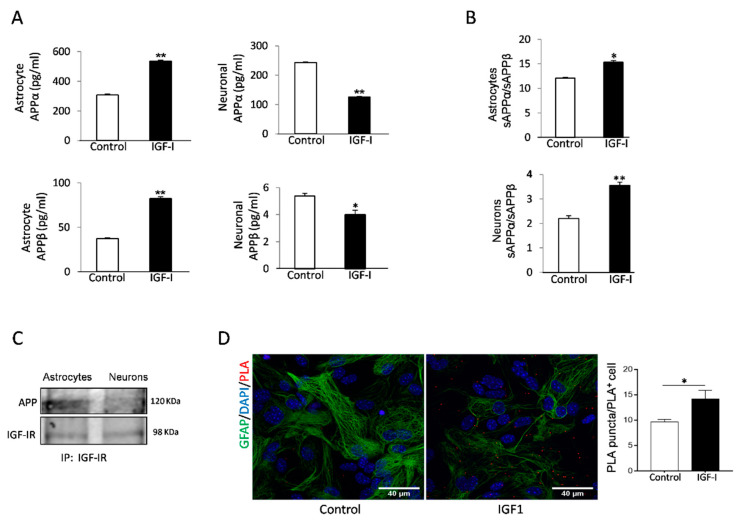
Aβ precursor protein (APP) processing in astrocytes is modulated by IGF-I. (**A**) IGF-I stimulated the secretion of both sAPPα and sAPPβ in cultured astrocytes (left histograms) while inhibited it in neurons (right histograms, *n* = 4). (**B**) However, IGF-I increased the sAPPα/sAPPβ ratio in both cell types, indicating a net non-amyloidogenic action of IGF-I in these cells. (**C**) IGF-IR and APP co-immunoprecipitate in cultured astrocytes, while in neurons, the interaction is negligible. (**D**) Proximity ligation assays (PLA) of APP and IGF-IR in cultured astrocytes confirm the interaction of both proteins that are upregulated by IGF-I (*n* = 3). Cell nuclei stained with Hoescht. * *p* < 0.05 and ** *p* < 0.01.

**Figure 4 ijms-21-09675-f004:**
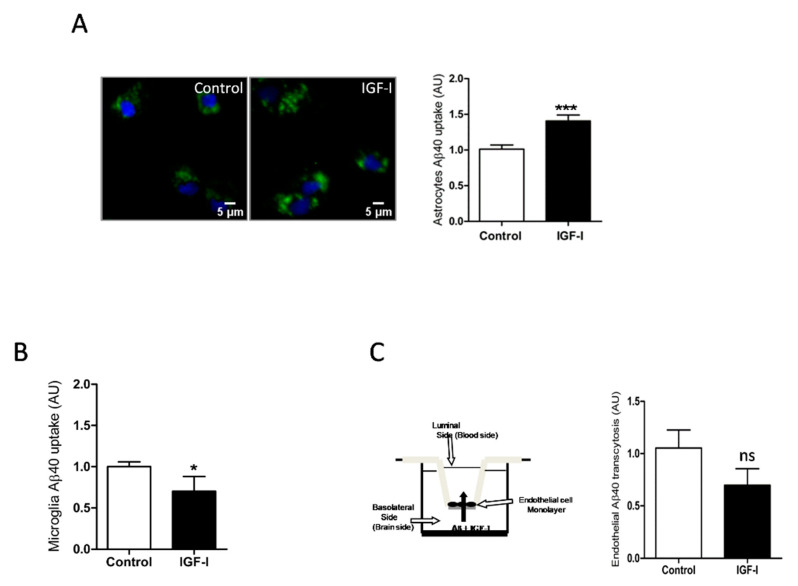
IGF-I modulates brain Aβ uptake in a cell-specific manner. (**A**) Aβ uptake by astrocytes is significantly increased by IGF-I (*n* = 8). Representative photomicrograph showing uptake by cultured astrocytes of fluorescently labeled Aβ (green). Cell nuclei stained with Hoescht. (**B**) Aβ uptake by microglia is significantly reduced by IGF-I (*n* = 7). (**C**) IGF-I did not significantly affect brain-to-blood efflux in an in vitro system mimicking the blood-brain-barrier (cartoon in the left). The amount of Aβ in the upper chamber was quantified 15 h after adding it to the lower chamber in the presence or absence of IGF-I (*n* = 6). * *p* < 0.05, *** *p* < 0.001 and not-significant (ns).

**Figure 5 ijms-21-09675-f005:**
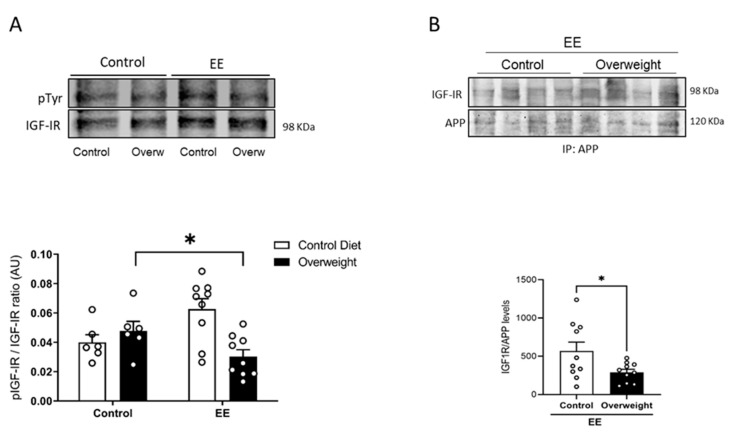
Reduced entrance of serum IGF-I in overweight mice. (**A**) In response to environmental enrichment (EE), overweight mice show lower brain IGF-IR phosphorylation than lean mice receiving a standard diet (*n* = 10 EE/6 Control, for each diet). (**B**) Interaction of APP with IGF-IR in the brain of mice submitted to EE stimulation was significantly decreased in overweight mice (*n* = 10 per group). Representative blots are shown, together with quantification histograms. * *p* < 0.05.

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
