# Peer review of "Circulating Insulin-Like Growth Factor I is Involved in the Effect of High Fat Diet on Peripheral Amyloid β Clearance"

_ijms, 2020, doi:10.3390/ijms21249675_

Round 1
Reviewer 1 Report
The Authors described the influence of high fat diet on the peripheral clearance of Abeta1-40 in overweight mice, demonstrating Aβ accumulation in hepatocytes, but unaltered brain Aβ levels and the involvement of IGF-I in enhancing peripheral Aβ clearance in overweight mice. The Authors performed valuable experiments in vitro and in vivo, in both naive and liver IGF-I 69 deficient mice, obtaining significant results.
The paper is original and interesting; however, I have some comments:
Major points:
- The Authors indicate the diet as a key factor in influencing peripheral clearance of Abeta, however it should be specified “high fat diet” since the crucial point is not the administration of a nutritional factor but the obesity, with the obesity-associated inflammation. Please modify this point.
- To have a complete picture of the alterations in Abeta and IGF-I levels between lean and overweight mice, I suggest to modify Figure 1, by adding Figure 5A. It should give a clearer indication that both Abeta and IGF-I levels are not altered in overweight mice brain, despite the significant alterations in the other compartments.
- The Authors suppose a reduced entrance of serum IGF-I in overweight mice brain following systemic IGF-1 administration on the base of the reduced brain IGF-1R/APP interaction. I wonder why they did not measure the brain IGF-I levels by ELISA. It should be very important to establish how much IGF-I really enters into the brain.
- In vitro results demonstrated that IGF-I increases the sAPPa/sAPPb ratio in both astrocytes and neurons, indicating a net non-amyloidogenic action of IGF-I. Do the Authors have any direct evidence of the sAPPa/sAPPb ratio in vivo, following systemic IGF-I treatment?
Author Response
- The Authors indicate the diet as a key factor in influencing peripheral clearance of Abeta, however it should be specified “high fat diet” since the crucial point is not the administration of a nutritional factor but the obesity, with the obesity-associated inflammation. Please modify this point.
Answer: Modified as indicated by reviewer. Please see title (ln 3), and text (ln: 28, 205, 223, 312 and 404).
- To have a complete picture of the alterations in Abeta and IGF-I levels between lean and overweight mice, I suggest to modify Figure 1, by adding Figure 5A. It should give a clearer indication that both Abeta and IGF-I levels are not altered in overweight mice brain, despite the significant alterations in the other compartments.
Answer: Figure 1 has been modified according to the indications of the reviewer. Please see new Fig 1 and its legend and displaced text in ln 218-221 and 228-29.
- The Authors suppose a reduced entrance of serum IGF-I in overweight mice brain following systemic IGF-1 administration on the base of the reduced brain IGF-1R/APP interaction. I wonder why they did not measure the brain IGF-I levels by ELISA. It should be very important to establish how much IGF-I really enters into the brain.
Answer: Thank you for the comment. We fully agree with the reviewer, but technical limitations preclude measuring injected IGF-I in brain. The amount of recombinant human IGF-I that enters the brain after peripheral injection is low and difficult to reliably measure with available techniques: ELISA lacks sensitivity and WB is unable to distinguish with certainty with endogenous murine IGF-I. We would very much like to be able to detect it, but previous trials were unsatisfactory. Formerly, we measured it using an RIA with an in-house produced anti-IGF-I antibody. However, many researchers asked us to give this antibody away and we run out of it. We have not produced new high-affinity antibodies since then, as commercial ELISAs become available.
- In vitro results demonstrated that IGF-I increases the sAPPa/sAPPb ratio in both astrocytes and neurons, indicating a net non-amyloidogenic action of IGF-I. Do the Authors have any direct evidence of the sAPPa/sAPPb ratio in vivo, following systemic IGF-I treatment?
Answer: No, we do not, and thank you for the nice idea. We will pursue it in on-going experiments that we need to perform to establish the correct experimental conditions to be able to detect both metabolites. We have added a sentence addressing this idea: ln 357-358.
Reviewer 2 Report
The by Herrero-Labrador et al., is well conducted, well design and also well written. The English is fluid and it is very clear explaining the different experiments. The experimental setting is also well designed. The paper now is good, but not completely novel and lack completely of patient data. Nevertheless, I think that the study have a great potential and coud be improved with some other experiments or analysis:
- AD was also correlated to inflammation. Do the authors checked if IGF-1 fluctuation impact on systemic inflammation sucg as cytokines, change in WBC and different lymphocytes subpopulations?
- Does the monocytes-macrophage compartment change its functionality upon IGF-1 treatment or inhibition?
- The last thing but more important at my eyes would be to check if in the authors hospitals or in the literature there some publicly available datasets of RNA expression levels to interrogate. It would be interesting if some expression data would be inferred and to see if IGF-1 could predict evolution of AD or are able to characterize a peculiar subgroup of patients.
- Please describe the role of IGF-1 with other form of dementia and, if possible, try to investigate its expression as above in other setting. It would be very interesting to know if IGF-1 has a role specifically in AD or on other type of dementia.
Author Response
AD was also correlated to inflammation. Do the authors checked if IGF-1 fluctuation impact on systemic inflammation sucg as cytokines, change in WBC and different lymphocytes subpopulations?
Answer: Thank you for the interesting comment. We have not yet gathered evidence of a link between changes in serum IGF-I and circulating cytokines. We are currently seeking funding for such a study to be done in parallel in mice models and human samples. As for WBC and lymphocytes we were not thinking to incorporate them but following your suggestion we will include these parameters as well.
Does the monocytes-macrophage compartment change its functionality upon IGF-1 treatment or inhibition?
Answer: Again, thank you for the comment. We do not work in the “immune side” of IGF-I function, although there are very interesting observations indicating an important role of this hormone in immune function. Following this suggestion, we will interact with immunologists to incorporate this interesting view in future studies in mice treated with IGF-I.
The last thing but more important at my eyes would be to check if in the authors hospitals or in the literature there some publicly available datasets of RNA expression levels to interrogate. It would be interesting if some expression data would be inferred and to see if IGF-1 could predict evolution of AD or are able to characterize a peculiar subgroup of patients.
Answer: Yes, indeed. Available information is abundant. There are many studies, including meta-analysis, and the current consensus is that more work is needed to reach firm conclusions on the role played by circulating IGF-I in relation to AD risk, evolution and so for. We are actively pursuing this idea.
Please describe the role of IGF-1 with other form of dementia and, if possible, try to investigate its expression as above in other setting. It would be very interesting to know if IGF-1 has a role specifically in AD or on other type of dementia.
Answer: This comment is related to the previous one. We have incorporated in the discussion new sentences commenting what is so far known about the role of IGF-I in AD and other dementias. See ln 335-39. This is a fluid topic, with new observations constantly accruing.
Reviewer 3 Report
This manuscript by Herrero-Labrador et al., reported the effect of HFD on peripheral and central Abeta metabolism, which appears to be dependent on IGF-1. Though results are potentially interesting, their main target is endogenous Abeta in wild type mice, which is often difficult to be measured. Thus, appropriate control (e.g., APP KO mice) should be shown or to be cited to support that their method can really address change of endogenous Abeta in wild-type mice.
Other points;
- Though they measured fluorescent signals in both in vivo and in cell experiments, it should be confirmed in part that these signals are from intact Abeta, or just fluorescent dye detached from degraded Abeta. Injected Abeta is rapidly degraded in blood..
- Regarding in vivo experiments, individual variation is not clear from the figures. Data of all mice should be shown as individual points in graph presentations (Fig. 1, 2 & 5), and all individual bands in western blotting (Fig. 5).
- It is not clear how Abeta was extracted from brains. Extraction method should be described in Material and Methods.
- In Figure 3C, APP band can be seen while authors stated band was negligible. Appropriate control (IPed with normal IgG, and non-IPed sample etc.) should be shown.
- Detail comparison with previous papers by the same group is important to clear why discrepant results were obtained. In this line, the current study did not use APP transgenic mice. Authors should more discuss potential difference of results if they use APP transgenic mice in the current study.
Author Response
Answers are enclosed in attached file

Round 2
Reviewer 1 Report
The manuscript has been significantly improved
Author Response
Thank you for your positive comment.
Reviewer 2 Report
The Authors,
have only partially answered to my comments but even if they could be complementary to the ongoing story I think that the Could use these For a follow-up story. Them, the authors have done a good literature research reasuming the state-of-art of the knowledge on IGF1 and AD. I think this topic Could be expanded in the future but I think For this paper it Could be enough to support the conclusions.
Author Response
Thank you for your positive comments. We certainly expect to expand our research on this topic incorporating your suggestions in relation to the impact on immune function
Reviewer 3 Report
This manuscript by Herrero-Labrador has revised the original manuscript according to reviewers’ comments. Although most points have been improved, the results of western blotting (Fig. 5) still do not clear my concerns. Based on the raw data of all individual blots for Fig. 5A in the response letter, it looks very weird about how the data of different blot is integrated into one analysis: although the comparison of bands in different blots is usually not possible without the common standard, how did authors calculate the densitometry, ratio, and combine mice from different blots? Also, in Figure 5C, it is strange that authors described that IGF-IR levels are increased by IGF-1 while no difference seems to exist (especially, when considered the ratio of IGF-IR/APP). Moreover, Figure 5C, bands of each protein (IGF-IR and APP) of control and overweight group should be shown in the continuous membrane without any separators. There is no description of Figure 5A in the text.
Author Response
Answer: Thank you for your comments. We performed the analysis of the samples shown in Figure 5A as follows. Densitometry was calculated with the software Image Studio Lite 5.2 (LI-COR) that automatically calculates the area of the band and subtracts background. This value is the raw data with which we calculate the ratio of the PTyr band with its corresponding IGF-IR band. In this way, we can compare ratios from different blots as normalization is done by total IGF-IR in each lane. Except for one blot that we only included animals from the EE group (because we had to introduce an additional batch of EE animals to complete lost samples), the rest of the blots include both the Control and EE groups. For the control diet group we find a mean ratio of all the animals in that group under control and EE conditions, and the same is true for the overweight mice. The reason we did not include a standard sample in all blots is that determination of pTyr IGF-IR by IP in brain requires a relatively large amount of tissue (particularly in non-stimulated, intact mice), which would mean sacrificing a batch of animals, pool their brains, and used the pool as inter-blot standard. Since the pTyr/IGF-IR ratio is a way to normalize each sample independently of other samples, we consider that we could minimize the use of animals following animal handling guidelines. We hope the reviewer considers that since raw values were obtained using a software independent of the experimenter bias, ratios of these raw values are a reliable way to normalize samples from different blots.
Following the comment of the reviewer, former Figure 5C has been eliminated. We did not calculate APP/IGF-IR ratios for it, so we cannot say whether there was an increase after IGF-I. See ln 298-99.
New Figure 5B: we include a continuous membrane as indicated by the reviewer.
We forgot to indicate Figure 5A in the revised text. We have corrected our mistake. See ln 297